# Association of Immune Semaphorins with COVID-19 Severity and Outcomes

**DOI:** 10.3390/biomedicines11102786

**Published:** 2023-10-13

**Authors:** Martina Vargovic, Neven Papic, Lara Samadan, Mirjana Balen Topic, Adriana Vince

**Affiliations:** 1Department for Infections in the Immunocompromised, University Hospital for Infectious Diseases, 10000 Zagreb, Croatia; mvargovic@bfm.hr; 2Department for Viral Hepatitis, University Hospital for Infectious Diseases, 10000 Zagreb, Croatia; avince@bfm.hr; 3School of Medicine, University of Zagreb, 10000 Zagreb, Croatia; lsamadan@mef.hr (L.S.); mbalen@bfm.hr (M.B.T.); 4Department for Gastrointestinal Infections, University Hospital for Infectious Diseases, 10000 Zagreb, Croatia

**Keywords:** COVID-19, SARS-CoV-2, semaphorins, SEMA3A, SEMA3F, SEMA3C, SEMA7A, ARDS, biomarkers

## Abstract

Semaphorins have recently been recognized as crucial modulators of immune responses. In the pathogenesis of COVID-19, the activation of immune responses is the key factor in the development of severe disease. This study aimed to determine the association of serum semaphorin concentrations with COVID-19 severity and outcomes. Serum semaphorin concentrations (SEMA3A, -3C, -3F, -4D, -7A) were measured in 80 hospitalized adult patients with COVID-19 (moderate (n = 24), severe (n = 32), critical, (n = 24)) and 40 healthy controls. While SEMA3C, SEMA3F and SEMA7A serum concentrations were significantly higher in patients with COVID-19, SEMA3A was significantly lower. Furthermore, SEMA3A and SEMA3C decreased with COVID-19 severity, while SEMA3F and SEMA7A increased. SEMA4D showed no correlation with disease severity. Serum semaphorin levels show better predictive values than CRP, IL-6 and LDH for differentiating critical from moderate/severe COVID-19. SEMA3F and SEMA7A serum concentrations were associated with the time to recovery, requirement of invasive mechanical ventilation, development of pulmonary thrombosis and nosocomial infections, as well as with in-hospital mortality. In conclusion, we provide the first evidence that SEMA3A, SEMA3C, SEMA3F and SEMA7A can be considered as new biomarkers of COVID-19 severity.

## 1. Introduction

COVID-19 presents with a wide range of clinical manifestations, from asymptomatic infection to pneumonia with acute respiratory distress syndrome (ARDS) and multiple-organ failure [1]. The severity of disease appears to be primarily influenced by the host’s immune response, which is extremely heterogeneous and complex, involving both innate and adaptive immunity [2,3]. Understanding immunopathogenesis and finding new biomarkers for the development of severe COVID-19 is necessary since they could serve as target sites for immunomodulation, which is the current cornerstone of severe and critical COVID-19 treatment, however, with modest clinical responses [4].

Semaphorins (SEMA) are a large family of secreted and membrane-bound signaling proteins expressed in most tissues, and divided into eight subclasses based on their Sema domain located on the N-terminal region [5]. They exhibit their effects through binding to their receptors, plexins (PLXN) and neuropilins (NRP), which can be modified by a variety of co-receptors (such as cell adhesion molecules, receptor tyrosine kinases), and transmembrane semaphorins themselves [6]. Although the molecular mechanisms of semaphorin signaling are still far from clear, the semaphorin’s interaction with its receptor complex alters the cell cytoskeleton’s structure and cell adhesion, which regulates cellular morphology and motility, as reviewed in [6].

Semaphorins were initially discovered as axonal guidance molecules in the development of the nervous system [7]. Subsequent research revealed their involvement in a variety of other physiological and pathophysiological processes, such as vascular growth [8], gene expression reprogramming of cancer cells [9], tumor progression [7], allergic diseases [10], cardiovascular diseases [11], metabolic disorders [12] or liver diseases [13,14]. Recently, increased focus has been placed on “immune semaphorins” and their roles in regulating immune cell activation, differentiation, mobility and migration in autoimmune diseases [15]. Several studies have demonstrated the potential of semaphorins as diagnostic and therapeutic targets in immune-mediated diseases [8,10,16,17].

However, the role of semaphorins in the immunopathogenesis of infections remains to be elucidated. Studies on mouse models of sepsis showed increased concentrations of several semaphorins in serum and tissues, and a blockade of semaphorins or their receptors led to a reduction in tissue damage and better survival rates [18,19]. In contrast, an experimental study of LPS-induced ARDS revealed decreased SEMA3A concentrations in lung tissue, while SEMA3A overexpression led to less severe lung impairment [20]. Depending on the secreting cells and receptors involved, each SEMA has different and often opposite functions. Notably, there are no studies in humans.

Here, we hypothesize that semaphorin concentrations, due to their key function in controlling immune responses, correlate with COVID-19 severity and outcomes.

## 2. Materials and Methods

### 2.1. Study Design and Patients

This study was part of a prospective, non-interventional cohort study that included consecutively hospitalized adult patients with COVID-19 at the University Hospital for Infectious Diseases Zagreb (UHID) in Croatia between April and December 2021 (part of the COVID-FAT trail, NCT04982328). At that time, the Delta (B.1.617.2 and AY lineages) SARS-CoV-2 variant predominated in Croatia (data were taken from the ECDC database on SARS-CoV-2 variants) [21]. The delta SARS-CoV-2 variant was shown to cause more severe disease and an excessive number of younger people dying despite receiving vaccinations [22,23]. Eighty patients with COVID-19 were included, and these patients have not been reported in previous studies. The sample size was selected according to power analysis for the Kruskal–Wallis test to achieve an 80% chance of detecting a difference in median semaphorin concentrations at a 5% significance level.

All included patients had bilateral pulmonary infiltrates on chest images. Patients who had a concomitant bacterial infection at the time of admission were excluded, as were those who began corticosteroid or antiviral medication prior to enrollment. Active malignant disease, pregnancy and immunosuppression (disease and/or current medical therapy including corticosteroids) were other exclusion factors.

Forty healthy, SARS-CoV-2-RNA-negative, age- and sex-matched healthcare workers were included as controls.

All participants provided written informed consent. The study followed the Declaration of Helsinki’s ethical principles and was approved by the UHID Ethics Committee (code 01-673-4-2021).

### 2.2. COVID-19 Disease Severity Classification

According to the National Institute of Health, COVID-19 severity was classified based on clinical symptoms, the oxygen level at admission and level of care (pandemic department or intensive care unit) [24]. Briefly, the severity of COVID-19 was classified into three categories: moderate (bilateral pneumonia with SpO2 > 93% on room air), severe (bilateral pneumonia with SpO2 ≤ 93% on room air, dyspnea and/or tachypnea > 24/min), and critical (intensive care unit, ARDS criteria, high-flow nasal cannula oxygen therapy (HFNC), non-invasive (NIV)/invasive mechanical ventilation (IMV)) [24].

### 2.3. Data Collection

At admission, demographic and comorbidity data were collected, including the presence of cardiovascular disease (CVD), arterial hypertension, chronic pulmonary disease (asthma, chronic obstructive pulmonary disease), chronic renal failure (CRF), diabetes mellitus, dyslipidemia, gastritis or gastroesophageal reflux disease (GERD) and chronic medications. All patients had their body mass index (BMI) measured.

The following routine laboratory results were obtained at admission: C-reactive protein (CRP), procalcitonin (PCT), ferritin, white blood cell count (WBC), absolute neutrophil (ANC) and lymphocyte count (ALC), platelet count (Plt), bilirubin, aspartate aminotransferase (AST), alanine aminotransferase (ALT), blood urea nitrogen (BUN), serum creatinine, gamma-glutamyl transferase (GGT), lactate dehydrogenase (LDH), fibrinogen and D-dimer.

The disease severity scores on the admission of each patient were calculated for MEWS [25], SOFA [26], PSI [27] and 4C mortality score [28].

The patients were treated according to the standard of care at the time, which included anticoagulants, remdesivir, tocilizumab, baricitinib and dexamethasone, at the discretion of the supervising physician.

Clinical monitoring, including oxygen requirements, invasive and non-invasive ventilation and complications, were assessed daily and collected in a standardized form.

### 2.4. Measurement of Semaphorin Serum Concentrations

Semaphorins were quantified using standardized enzyme-linked immunosorbent assay (ELISA) (Human Semaphorin-3A, -3F, -4D and -7A by ELISA kit, AssayGenie, Dublin, Ireland, and Human Semaphorin-3C ELISA Kit, MyBioSource, San Diego, CA, USA), as suggested by the manufacturer.

### 2.5. Statistical Analysis

Clinical, laboratory and demographic data were analyzed and reported descriptively as frequencies and medians with interquartile ranges. To compare two groups, Fisher’s exact test and the Mann–Whitney U test were used. To compare three or more groups, the Kruskal–Wallis test with Dunn’s multiple comparisons test was used. All tests were two-tailed, with a statistically significant *p*-value of 0.05. Spearman’s rank correlation coefficient was used to examine correlations, which were then summarized in a correlation matrix. A receiver operating characteristic (ROC) analysis was used to compare the discriminatory performance of the laboratory variables under consideration. Time to hospital discharge or readiness for discharge stratified by biomarker levels was evaluated using the Kaplan–Meier method and hazard ratios (HR) with 95% confidence intervals (95% CI) and *p*-values were calculated by the log-rank test. Risk factors associated with critical COVID-19 were investigated using a univariate and subsequently multivariable logistic regression analysis. The strength of association was expressed as an odds ratio (OR) and its corresponding 95% CI. GraphPad Prism Software version 10 (San Diego, CA, USA) was used for statistical analyses.

## 3. Results

### 3.1. Baseline Patients’ Characteristics

Eighty patients with COVID-19 (44 males (55%); median age 62, IQR 47–68 years) and 40 controls (20 males (50%); median age 59, IQR 41–67 years) were included.

COVID-19 severity was categorized as moderate in 24 (30%), severe in 32 (40%), and critical in 24 (30%) patients. As shown in Table 1, there were no differences in the age, gender, comorbidities, chronic medications or duration of symptoms before admission between the groups. Patients with critical COVID-19 had a higher BMI than those with severe or moderate COVID-19. As expected, the severity of clinical symptoms and disease severity scores, including MEWS, SOFA, PSI and the 4C mortality score, differed significantly between groups (Table 1). Patients in the critical group had lower peripheral oxygen saturation (82%, IQR 78–87 vs. 88%, IQR 82–89 vs. 95% IQR 93–96, *p* = 0.0001) and a lower PaO2/FiO2 ratio (126 IQR 76–174 vs. 183 IQR 137–250 vs. 347 IQR 323–428, *p* = 0.0001) on admission.

As illustrated in Table 2, patients with critical and severe COVID-19 had significantly higher serum concentrations of CRP and IL-6, glucose, urea, AST, LDH and CK at the time of admission. There were no differences in other routine laboratory findings.

In the group of severe COVID-19, oxygen supplementation had a median of 9.5 L (IQR 4–15 L) and the duration of required oxygen supplementation was 6.5 days (IQR 4.7–9 days). In the critical group, all patients required HFNC (median duration of 6 days, IQR 3–8 days), 12 (50%) of them received NIV during hospitalization (median duration of 2 days (IQR 1–6 days). Ten patients (41.67%) required invasive mechanical ventilation (median duration of 4 days, IQR 3–11 days), 6 patients (25%) needed continuous renal replacement therapy (CRRT), and one patient was treated with veno-venous extracorporeal membrane oxygenation (vv-ECMO).

Overall, 47 (58.75%) patients were treated with remdesivir, 76 (95%) with dexamethasone, 12 (15%) with tocilizumab and 4 (5%) with baricitinib.

In 10 patients (12.5%), pulmonary thrombosis was diagnosed (1 in the mild, 4 in the severe and 5 in the critical group). Nosocomial infections were diagnosed in 16 patients (20%); 10 patients with critical, 5 with severe and 1 with mild COVID-19. Ten patients (12.5%) in our cohort died during hospitalization (7 males, median age of 63, IQR 50–68 years).

### 3.2. Analysis of Serum Semaphorin Concentrations in Patients with COVID-19 and Healthy Controls

Concentrations of the serum semaphorins SEMA3A, SEMA3C, SEMA3F, SEMA4D and SEMA7A were detectable in all patients with COVID-19 (Figure 1). Patients with COVID-19 had significantly higher SEMA3C (1.6 ng/mL, IQR 0.8–2.4 vs. 0.5 ng/mL, IQR 0.2–1.1, *p* ≤ 0.0001), SEMA3F (4.6 ng/mL, IQR 4.1–6.0 vs. 1.2 ng/mL, IQR 0.9–1.5, *p* ≤ 0.0001) and SEMA7A (1.5 ng/mL, IQR 0.7–2.7 vs. 0.9 ng/mL, IQR 0.2–1.3, *p* ≤ 0.0001) concentrations compared to healthy controls (Figure 1a, Table 3). In contrast, patients with COVID-19 had significantly lower SEMA3A concentrations (14 ng/mL, IQR 11–16 vs. 21 ng/mL, IQR 15–24, *p* ≤ 0.0001). There were no statistical differences in SEMA4D serum concentrations between groups (32 ng/mL, IQR 31–73 and 47 ng/mL, IQR 21–81, *p* = 0.4459).

Next, ROC analysis was performed to determine cut-off values of serum semaphorin concentrations for differentiating patients with COVID-19 from healthy controls (Figure 1b and Table 3). All tested semaphorins showed good accuracy in distinguishing patients with COVID-19 from HC. A cutoff value of SEMA3F > 2.10 ng/mL correctly predicted COVID-19 with a sensitivity of 97% and specificity of 97% (AUC 0.99, 95%CI 0.99–1.00). SEMA3A < 15.5 ng/mL predicted COVID-19 with a sensitivity of 71% and specificity of 72% (AUC 0.79, 95%CI 0.72–0.87). SEMA3C > 1.00 ng/mL showed a sensitivity of 63% and specificity of 66% (AUC 0.77, 95%CI 0.71–0.85) and SEMA7A >1.15 ng/mL had a sensitivity of 64% and specificity of 65% (AUC 0.73, 95%CI 0.66–0.81).

### 3.3. Correlation of Semaphorin Serum Concentrations with COVID-19 Severity

As shown in Figure 2 and Table 4, serum concentrations of SEMA3A and SEMA3C were negatively correlated with disease severity, with the lowest concentrations in the most severely ill patients. SEMA4D serum concentrations showed no correlation with disease severity. In contrast, serum concentrations of SEMA3F and SEMA7A positively correlated with COVID-19 severity, with the highest levels in patients with critical COVID-19 (Figure 2, Table 3).

### 3.4. ROC Analysis of Serum Semaphorins Concentrations in Predicting COVID-19 Severity

As shown in Table 4 and Figure 3a, ROC analysis was performed to determine the threshold value of serum semaphorin concentrations for differentiating patients with moderate from patients with severe/critical COVID-19. A cutoff value > 14 ng/mL of the SEMA3A correctly predicted moderate COVID-19 with a sensitivity of 62% and specificity of 66% (AUC 0.75, 95%CI 0.66–0.85). SEMA3F < 4.35 ng/mL had a sensitivity of 73% and specificity of 71% (AUC 0.74, 95%CI 0.54–0.84), SEMA7A < 1.23 ng/mL showed a sensitivity of 66% and specificity of 58% (AUC 0.71, 95%CI 0.62–0.80) and SEMA3C > 1.80 ng/mL showed a sensitivity of 60% and specificity of 58% (AUC 0.66, 95%CI 0.56–0.76). The diagnostic accuracy was similar to routinely measured CRP, IL-6 and LDH.

Next, ROC analysis was performed to determine the cut-off value of serum semaphorin concentrations for differentiating patients with critical COVID-19 from patients with moderate/severe COVID-19 (Figure 3b). A cutoff value < 13 ng/mL of the SEMA3A predicted critical COVID-19 with a sensitivity of 70% and specificity of 71% (AUC 0.78, 95%CI 0.69–0.87), SEMA3F > 4.7 ng/mL showed a sensitivity of 64% and specificity of 71% (AUC 0.76, 95%CI 0.66–0.85), SEMA7A > 2.0 ng/mL showed a sensitivity of 82% and specificity of 71% (AUC 0.82, 95%CI 0.74–0.89). Serum semaphorin levels show better predictive values than CRP, IL-6 and LDH for differentiating critical from moderate/severe COVID-19 (Table 4).

Finally, we performed a multivariable logistic regression analysis to identify factors associated with the development of critical COVID-19. After adjustment for potential cofounders, SEMA3F > 4.7 ng/mL (OR 5.73, 95%CI 1.38–29.76), SEMA7A > 2.0 ng/mL (OR 12.76, 95%CI 2.45- 96.32), admission paO2/FiO2 < 150 (OR 3.76, 95%CI 1.03–14.45) and 4C mortality score > 9 (OR 3.69, 95%CI 1.12–13.42) were associated with the critical disease, while age, sex, comorbidities, CCI, obesity and laboratory parameters such as CRP and LDH were not associated with critical disease in our model (AUC 0.89, 95%CI 0.81–0.96).

### 3.5. Association of Serum Semaphorin Concentrations with COVID-19 Clinical Outcomes and Complications

We examined the impact of serum semaphorin concentrations on time to recovery, as defined by time to hospital discharge or readiness for discharge by day 28. In a survival analysis using Kaplan–Meier estimates, SEMA3A < 13 ng/mL (HR 4.4, 95% CI 1.21–16.55, *p* = 0.0116), SEMA3F > 4.7 ng/mL (HR 5.24, 95% CI 1.38–19.88, *p* = 0.0186) and SEMA7A > 2.0 ng/mL (HR 10.11, 95% CI 2.55–40.01, *p* = 0.0002) appeared to be an efficient prognostic biomarker associated with time to recovery.

Twelve patients required mechanical ventilation (including NIV and IMV) during hospitalization. In this group, SEMA3F (6.0 ng/mL, IQR 4.5–6.7 vs. 4.5 ng/mL, IQR 3.9–5.4, *p* = 0.0308) and SEMA7A (2.8 ng/mL, IQR 2.2–5.8 vs. 1.3 ng/mL, IQR 0.55–1.9, *p* = 0.0004) were significantly higher than in patients who did not require advanced respiratory support.

Ten patients were diagnosed with pulmonary thrombosis, and these patients had significantly higher serum concentrations of SEMA3C (2.9 ng/mL, IQR 2.3–3.6 vs. 1.4 ng/mL, IQR 0.84–2.3, *p* = 0.0007), SEMA3F (6.5 ng/mL, IQR 4.7–6.6 vs. 4.4 ng/mL, IQR 4.0–5.7, *p* = 0.0339) and SEMA7A (2.9 ng/mL, IQR 1.2–7.1 vs. 1.4 ng/mL, IQR 0.76–2.2, *p* = 0.0471).

Next, we analyzed baseline serum semaphorin concentrations with the development of nosocomial infections as a complication of COVID-19. Patients who developed nosocomial infections had significantly higher levels of SEMA3F (6.0 ng/mL, IQR 4.5–6.4 vs. 4.5 ng/mL, IQR 3.9–5.6, *p* = 0.0411) and SEMA7A (2.8 ng/mL, IQR 1.90–4.2 vs. 1.3 ng/mL, IQR 0.59–1.9, *p* = 0.0005).

Overall, 10 patients in our cohort died during hospitalization. Non-survivors had significantly lower SEMA3A (11 ng/mL, IQR 6.9–14 vs. 13 ng/mL, IQR 11–16, *p* = 0.0269) and higher SEMA7A (2.8 ng/mL, IQR 2.0–8.6 vs. 1.4 ng/mL, IQR 0.59–2.0, *p* = 0.0029) serum concentrations.

### 3.6. Correlation Analysis of Serum Semaphorin Concentrations with Routine Clinical and Laboratory Parameters

We analyzed potential correlations among paired laboratory parameters, including semaphorin concentrations and clinical variables in patients with COVID-19, as presented in Figure 4. Serum SEMA3A negatively correlated with the 4C mortality score (r = −0.22, *p*= 0.05), AST (r = −0.25, *p* = 0.03), LDH (r = −0.36, *p* < 0.01) and CK (r = −0.27, *p* = 0.02). Serum SEMA3C positively correlated with SEMA4D (r = 0.32, *p* = 0.01), ANC (r = 0.23, *p* = 0.05), and negatively with CK (r = −0.25, *p* = 0.03). Serum SEMA3F positively correlated with SEMA7A (r = 0.39, *p* < 0.01), 4C mortality score (r = 0.30, *p* = 0.01), Hb (r = 0.32, *p* < 0.01), urea (r = 0.25, *p* = 0.02), LDH (r = 0.32, *p* < 0.01) and d-dimer (r = 0.26, *p* = 0.02), while negatively with ALP (r = −0.23, *p* = 0.04). Serum SEMA4D positively correlated with SEMA3C (r = 0.32, *p* = 0.01). Serum SEMA7A positively correlated with SEMA3F (r = 0.39, *p* < 0.01), 4C mortality score (r = 0.44, *p* < 0.01), CRP (r = 0.35, *p* < 0.01), procalcitonin ( r = 0.27, *p* = 0.02), ANC (r = 0.42, *p* < 0.01), Plt (r = 0.41, *p* < 0.01), urea (r = 0.41, *p* < 0.01), LDH (r = 0.25, *p* = 0.02), Troponin T (r= 0.31, *p* = 0.01), fibrinogen ( r = 0.36, *p* < 0.01), d-dimer (r = 0.21, *p* = 0.05), and negatively correlated with albumin level( r = −0.25, *p* = 0.03).

## 4. Discussion

In this study, we provide the first evidence that COVID-19 patients have different semaphorin serum concentrations as compared to healthy controls. While SEMA3A was decreased, SEMA3C, SEMA3F and SEMA7A were increased in COVID-19. Furthermore, we showed an association of semaphorin levels with COVID-19 severity; SEMA3F and SEMA7A were higher in critical COVID-19, while SEMA3A and SEMA3C negatively correlated with COVID-19 severity and were lower in the critical group. Semaphorins showed equal or better accuracy in predicting disease severity than the widely used CRP, IL-6 or LDH.

Firstly, we found decreased serum concentrations of SEMA3A that further decreased with COVID-19 severity. Class 3 semaphorins, specifically SEMA3A, have immunosuppressive and regulatory effects that include neutrophil migration, induce the shift of activated macrophages (M1) to the resolution phase phenotype (M2) and negatively control the T cell-mediated response predominantly via the activation of regulatory T cells (Tregs) [29]. In patients with autoimmune diseases (e.g., SLE, rheumatoid arthritis, systemic sclerosis and allergic diseases), a reduced expression of SEMA3A correlated with T-cell-mediated inflammation and disease severity [10,15]. The role of SEMA3A in the pathogenesis of infections might depend on the receptor utilized, which has been the subject of several experimental studies. Inhibition of the SEMA3A/PLXNA4 complex attenuates Toll-like receptor (TLR) pathways, which was associated with a decreased septic response and improved survival rates [30]. In contrast, inhibition of the SEMA3A/NRP-1 complex demonstrated increased production of proinflammatory cytokines and higher mortality [31]. In the transcriptome study (GEO dataset GSE57011), SEMA3A was identified as the most downregulated gene in ARDS patients [20], and overexpression of SEMA3A in the lipopolysaccharides (LPS)-induced ARDS model alleviates oxidative stress and inflammation by suppressing activation of the extracellular signal-regulated kinase/Jun-N-Terminal Kinase (ERK/JNK) signaling pathway in rat pulmonary microvascular endothelial cells [20].

Since the fine-tuned immune response is vital in determining the outcome of the SARS-CoV-2 infection [32], we can hypothesize that decreased serum concentrations of SEMA3A in COVID-19 patients and its negative correlation with disease severity might result in the lack of anti-inflammatory and immunosuppressive effects of SEMA3A, which might lead to an uncontrolled inflammatory cascade.

Next, we found that SEMA3C is increased in moderate, but not in critical, COVID-19. Furthermore, SEMA3C was increased in a subgroup of patients diagnosed with pulmonary thrombosis. Less is known about the immunoregulatory role of SEMA3C, but it was shown that SEMA3C regulates fibrosis, vascular development, pathological angiogenesis and the migration of tumor cells. The expression of SEMA3C is related to tumor progression and poor prognosis in lung cancer, prostate, breast cancer, gastric cancer and ovarian cancer, which makes it a potential therapeutic target for malignant diseases [33,34]. SEMA3C regulates extracellular matrix composition through increased expression of IL-6, transforming growth factor-β (TGF-β) and connective tissue growth factor (CTGF), and was implicated in the development of liver fibrosis [13,14,35]. Recently, it was shown that SEMA3C plays an important role in the development of murine acute kidney injury by promoting vascular permeability, interstitial edema, leukocyte infiltration and tubular injury [36]. After administration of SEMA3C in murine models, systemic and renal hemodynamics changed: mean arterial pressure decreased and vascular resistance was reduced [36]. SEMA3C has a pivotal role in vascular smooth muscle cell migration and cardiovascular system development [37,38]. Interestingly, SEMA3C might regulate pathological angiogenesis, where SEMA3C exerted potent inhibiting effects and was suggested as a potent and selective inhibitor of pathological retinal angiogenesis [39]. Since severe COVID-19 is characterized by significant distortion of the lung angioarchitecture, small vessel vasculitis and microthrombosis, all associated with worse prognosis and increased mortality [40], we can theorize that SEMA3C might play a role in aberrant angiogenesis associated with the development of severe ARDS that is still to be investigated.

Similarly, SEMA3F and SEMA7A concentrations were increased and positively correlated with COVID-19 disease severity and were identified as predictors of COVID-19 outcomes, including the need for mechanical ventilation, development of pulmonary thrombosis and nosocomial infections. Both SEMA3F and SEMA7A are secreted by activated immune cells and have emerged as regulators of neutrophil migration, vascular permeability and cytoskeletal remodeling, and the initiation of an inflammatory signaling cascade in models of acute lung injury. An increased number of neutrophils in bronchoalveolar fluid with a high expression of SEMA3F and NRP2 was observed in a murine model after the LPS challenge, and neutrophil-specific loss of SEMA3F resulted in more rapid neutrophil recruitment and clearance from the lungs [41]. By interacting with various receptors and organ systems, SEMA7A has opposing effects: in interactions with integrin receptors, SEMA7A has a protective and anti-inflammatory effect, while via the plexin C1 receptor it promotes extravascular neutrophil migration and the release of pro-inflammatory cytokines [42]. In the murine model, SEMA7A causes transendothelial migration of neutrophils into lung tissue [43,44], and a blockade of SEMA7A in in vivo and in vitro models showed reduced injury-induced neutrophil influx, correlating with reduced lung injury along with reduced cytokine response [18]. To summarize, the SEMA3F and SEMA7A in COVID-19 might regulate vascular permeability and cytoskeletal remodeling along with neutrophil migration and retention in inflamed tissue, which leads to the amplification of inflammation. 

In our study, there were no differences in SEMA4D serum concentrations between healthy controls and COVID-19 patients. SEMA4D regulates immune activation and inflammatory responses by modulating cytoskeleton reorganization through its principal receptor, CD72, located on T cells, B cells, macrophages and dendritic cells [45]. In animal models of autoimmune diseases such as multiple sclerosis and autoimmune encephalomyelitis, SEMA4D correlated with disease severity [46], as in patients with psoriasis and rheumatoid arthritis [47]. SEMA4D is most extensively studied in oncology and it is currently considered a promising target for antitumor therapy for breast cancer [48]. The impact of class IV semaphorins on COVID-19 outcomes should be further examined.

Recent research has shown that micro RNAs (miRNA) control SEMA signaling in immune, cardiovascular and nervous systems, and malignancies, directly through their receptors, or indirectly by modulating the molecules that regulate the expression of semaphorins [49]. Similarly, there are reports that specific miRNA signatures in blood or respiratory samples can distinguish COVID-19 disease severity or COVID-19 patients from healthy people [50]. Some of them are linked with semaphorin signaling such as miR-17-5p [51], miR-142-5p [52], miR-126-3p [53], miR-19b-3p [54], miR-92a-3p [55] and miR-320a [56], thus highlighting the potential importance of semaphorin signaling in COVID-19. Interestingly, SARS-CoV-2-encoded miRNAs were shown not only to target viral genomes and alter viral fitness, but can also be transported to host cells during viral infection and bind to host miRNAs and genes and alter immune responses [57]. However, their role in regulating semaphorins remains to be elucidated.

This study should be viewed within its limitations; since this was an observational study, causality could not be determined; a relatively small number of participants in COVID-19 severity subgroups limits statistical analysis and should be confirmed in a larger population; the impact of comorbidities on semaphorins and other inflammatory markers was not evaluated; the concentrations of semaphorins were determined at a single time point, and dynamic variations related to clinical outcomes were not examined. Nevertheless, we studied a well-defined cohort of patients and report the first data examining the semaphorins’ profile in patients with COVID-19. Additional studies are needed for a better understanding of the complex underlying immunopathological mechanisms including their effect on development and activation of B and T cells, and how semaphorins contribute to COVID-19 progression.

## 5. Conclusions

In conclusion, we have shown that patients with COVID-19 have different expressions of SEMA3A, SEMA3C, SEMA3F and SEMA7A than healthy controls, which correlate with disease severity and outcomes. Due to their role in regulating inflammation, cell migration, fibrosis and angiogenesis, which have already been explored in neoplastic and autoimmune diseases, semaphorins could be new diagnostic and prognostic biomarkers and potential therapeutic targets in COVID-19, which warrant further investigation.

## Figures and Tables

**Figure 1 biomedicines-11-02786-f001:**
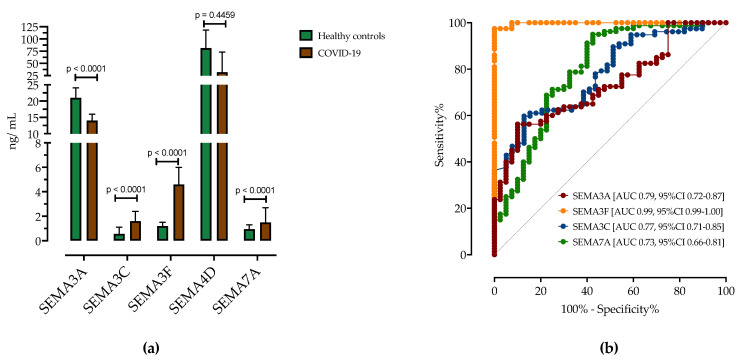
(**a**) Serum concentrations of semaphorin measured by ELISA in healthy controls and in patients with COVID-19. (**b**) ROC curve analysis of serum semaphorins for determination of COVID-19. Shown are AUCs with corresponding 95% CI.

**Figure 2 biomedicines-11-02786-f002:**
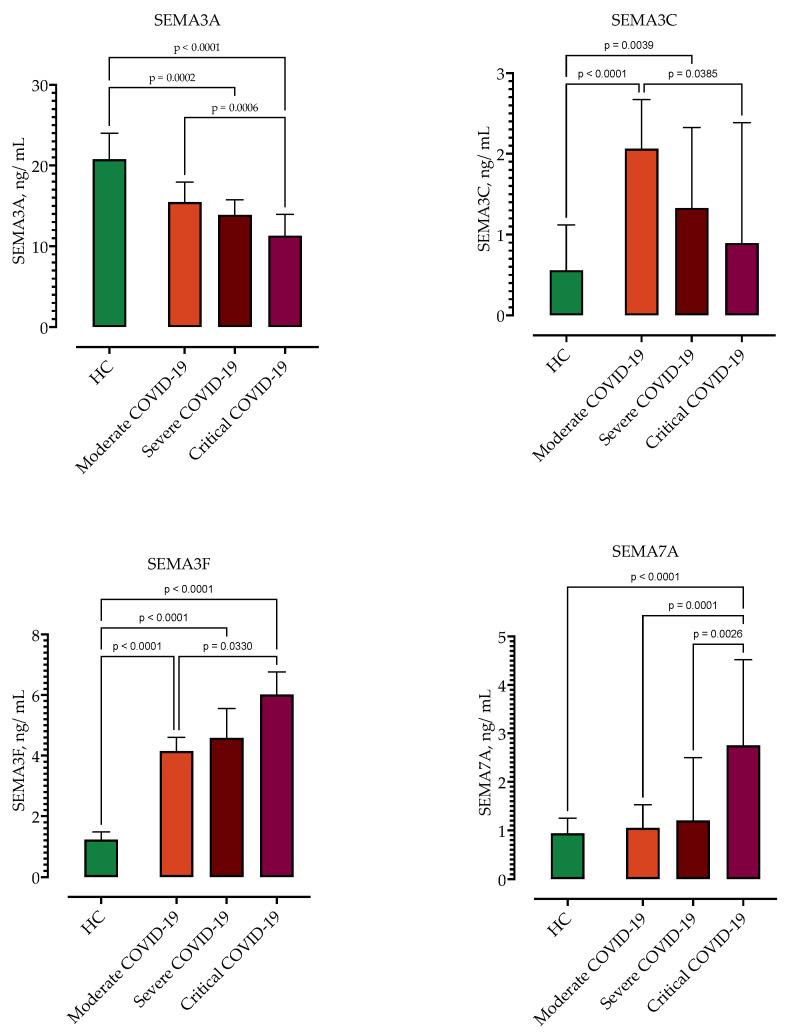
Serum concentrations of semaphorin SEMA3A, SEMA3C, SEMA3F and SEMA7A in healthy controls (HC) and patients with COVID-19 stratified by disease severity (moderate, severe, critical). Data are presented as medians with interquartile ranges. The *p*-values are calculated by Kruskal–Wallis test with Dunn’s multiple comparisons test.

**Figure 3 biomedicines-11-02786-f003:**
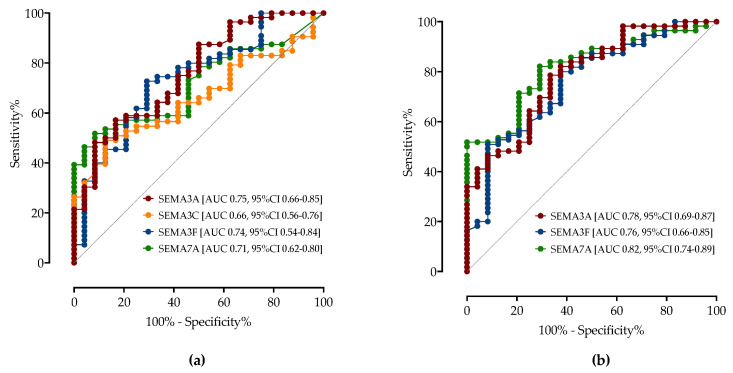
The ROC curve analysis of serum semaphorin concentrations for discrimination of (**a**) moderate COVID-19 and severe/critical COVID-19 patients; (**b**) critical COVID-19 and moderate/severe COVID-19. Shown are AUCs with 95% CI.

**Figure 4 biomedicines-11-02786-f004:**
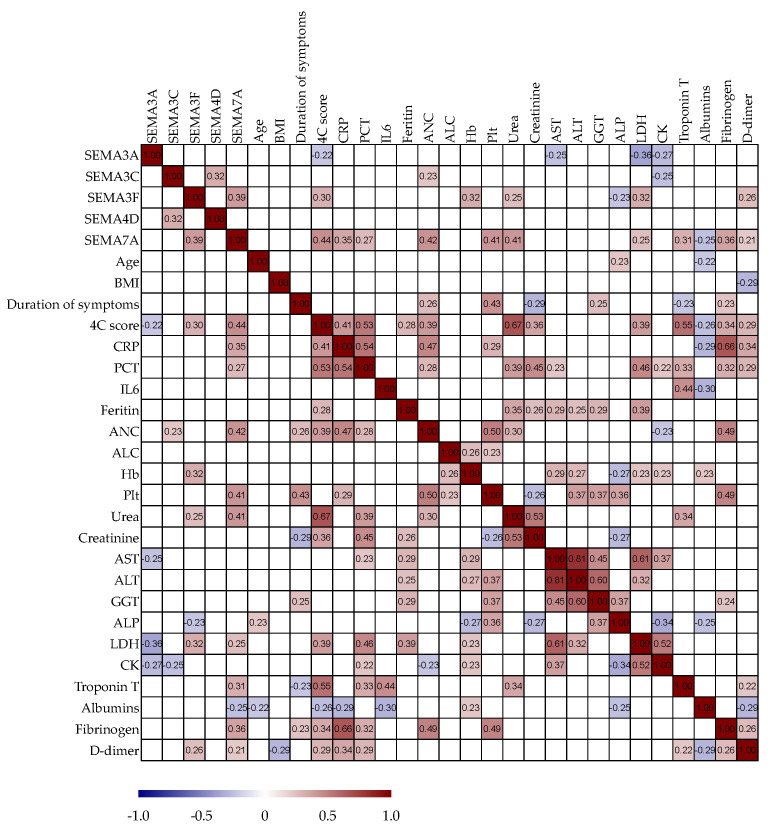
Spearman correlation correlogram. The strength of the correlation between two variables is represented by the color at the intersection of those variables. Colors range from dark blue (strong negative correlation; r = −1.0) to red (strong positive correlation; r = 1.0). Results were not displayed if *p* > 0.05.

**Table 1 biomedicines-11-02786-t001:** Baseline patients’ characteristics.

	Moderate n = 24	Severe n = 32	Critical n = 24	*p*-Value
Male sex	13 (54.17%)	19 (59.38%)	12 (50%)	0.7801
Age, years	64 (48–69)	62 (44–71)	63 (48–65)	0.8829
BMI, kg/m^2^	26 (24–29)	29 (27–34)	30 (25–31)	0.0517
Waist–hip ratio (WHR)	0.97 (0.91–1)	1 (0.97–1.1)	0.98 (0.92–1.1)	0.0744
**Comorbidities**				
Charlson comorbidity index	1.5 (0–3.8)	2 (0.25–3)	2 (1.3–3.8)	0.5018
Diabetes mellitus	8 (33.33%)	4 (12.50%)	6 (25.00%)	0.1706
Arterial hypertension	13 (54.17%)	17 (53.13%)	11 (45.83%)	0.8152
COPD	2 (8.33%)	3 (9.38%)	2 (8.33%)	0.9870
Dyslipidemia	3 (12.50%)	7 (21.88%)	5 (20.83%)	0.6411
Gastritis/GERD	4 (16.67%	1 (3.13%)	2 (8.33%)	0.2062
Cardiovascular diseases	5 (20.83%)	4 (12.50%)	5 (20.83%)	0.6301
No comorbidities	7 (29.17%)	10 (31.25%)	6 (25.00%)	0.8761
**Chronic medications**				
ACE inhibitors	5 (20.83%)	15 (46.88%)	13 (54.17%)	0.0563
Acetylsalicylic acid	3 (12.50%)	4 (12.50%)	1 (4.17%)	0.5230
Beta blockers	4 (16.67%)	12 (37.50%)	9 (37.50%)	0.1832
Proton pump inhibitors	5 (20.83%)	4 (12.50%)	2 (8.33%)	0.4379
Hypolipidemic	3 (12.50%)	6 (18.75%)	5 (20.83%)	0.7279
Peroral hypoglycemics	5 (20.83%)	7 (21.88%)	6 (25.00%)	0.9364
No medications	7 (29.17%)	10 (31.25%)	6 (25.00%)	0.8761
**Disease severity at admission**				
Duration of illness, days	9 (7–9)	10 (7–11)	9 (7.3–9.8)	0.8483
Body temperature, °C	38 (37–38)	38 (37–38)	37 (37–38)	0.2829
Heart rate/min	93 (84–100)	96 (80–105)	98 (86–107)	0.5774
Mean arterial pressure, mmHg	94 (87–103)	93 (83–100)	100 (89–108)	0.1196
Respiratory rate/min	21 (18–24)	25 (21–30)	27 (22–31)	**0.0006**
SpO2 on room air, %	95 (93–96)	88 (82–89)	82 (78–87)	**<0.0001**
Pao2/FiO2 ratio	347 (323–428)	183 (137–250)	126 (76–174)	**<0.0001**
MEWS	2 (1–3.5)	3 (2–4)	3 (2–3.8)	**0.0422**
SOFA	1.5 (1–2)	2 (2–3)	2 (2–3)	**0.0008**
PSI	54 (41–73)	64 (52–77)	84 (58–99)	**0.0078**
4C mortality score	5 (3–8)	9 (6–11)	11 (8–12)	**<0.0001**

Data are presented as medians with interquartile ranges (IQR) or frequencies with percentages (n, %). Abbreviations: Body mass index (BMI); Charlson comorbidity index (CCI); Chronic Obstructive Pulmonary Disease (COPD); Gastro-esophageal reflux disease (GERD); Angiotensin-converting enzyme (ACE) inhibitors; Modified Early Warning Score (MEWS); Sequential Organ Failure Assessment (SOFA) Score; Pneumonia Severity Index (PSI); Coronavirus Clinical Characterization Consortium (4C) Mortality Score.

**Table 2 biomedicines-11-02786-t002:** Laboratory findings at admission.

	Moderate n = 24	Severe n = 32	Critical n = 24	*p*-Value
CRP, mg/L	83 (30–120)	101 (67–171)	133 (75–213)	0.0214
Procalcitonin, µg/L	0.11 (0.058–0.19)	0.13 (0.073–0.41)	0.2 (0.094–0.69)	0.0611
Interleukin-6, pg/mL	15 (6.5–54)	61 (18–107)	68 (18–142)	0.0102
Ferritin, µg/L	711 (443–1222)	841 (623–1468)	945 (656–2063)	0.2179
WBC, ×10^9^/L	5.9 (4.8–8.7)	6.3 (4.1–8.2)	6.4 (5.1–11)	0.3815
Lymphocyte count, 10^9^/L	0.83 (0.55–1.1)	0.6 (0.44–0.88)	0.68 (0.49–0.83)	0.3235
Neutrophil count, 10^9^/L	4.4 (3.5–6.4)	5.1 (3.2–6.7)	5.4 (4.2–9)	0.2769
Hemoglobin, g/L	135 (117–147)	134 (127–145)	140 (126–148)	0.5138
Platelets, ×10^9^/L	166 (120–277)	190 (137–247)	197 (141–246)	0.6331
Glucose, mmol/L	6.5 (5.8–7.9)	7.2 (6.6–8.1)	8.2 (6.9–9.8)	0.0367
Urea, mmol/L	4.4 (3.2–5.1)	6.2 (4.5–9.2)	6.1(4.7–9.3)	0.0003
Creatinine, µmol/L	72 (64–83)	80 (67–97)	69 (60–99)	0.2905
eGFR, ml/min/1.73 m^2^	99 (80–107)	89 (71–106)	91 (57–100)	0.1748
Bilirubin, µmol/L	10 (8–12)	13 (10–16)	11 (8.3–14)	0.1045
AST, IU/L	39 (26–51)	59 (34–86)	49 (37–90)	0.0104
ALT, IU/L	29 (21–57)	51 (28–75)	31 (24–65)	0.1682
GGT, IU/L	50 (30–55)	47 (26–124)	43 (2293)	0.5901
LDH, IU/L	262 (199–319)	382 (251–490)	483 (370–598)	<0.0001
CK, IU/L	104 (56–272)	173 (66–319)	269 (120–524)	0.0352
Albumins, g/L	38 (36–41)	37 (35–41)	37 (35–39)	0.2015
Fibrinogen, g/L	5.6 (5–6.4)	6.2 (5.4–6.9)	5.8 (5.4–6.9)	0.3437
D-dimer, mg/L	0.95 (0.63–2)	0.86 (0.5–1.6)	1 (0.52–1.3)	0.7481

Data are presented as medians with interquartile ranges (IQR), *p*-values are calculated by Kruskal–Wallis test. Abbreviations: C-reactive protein (CRP); White blood cell Count (WBC); Blood urea nitrogen (BUN); estimated glomerular filtration rate (eGFR); Aspartate Aminotransferase (AST); Alanine Aminotransferase (ALT); Gamma-glutamyl transferase (GGT); Lactate dehydrogenase (LDH); Creatinine Kinase (CK).

**Table 3 biomedicines-11-02786-t003:** Serum concentrations of semaphorins in healthy controls and COVID-19 patients and ROC analysis of sensitivity and specificity in differentiating COVID-19 patients from healthy controls.

	**Healthy Controls (n = 40)**	**COVID-19 Patients (n = 80)**	**Difference (95% CI)**	***p*-Value**
SEMA3A, ng/mL	21 (15–24)	14 (11–16)	−7.2 (−8.6–−4.2)	<0.0001
SEMA3C, ng/mL	0.56 (0.25–1.1)	1.6 (0.84–2.4)	1.1 (0.4–1.2)	<0.0001
SEMA3F, ng/mL	1.2 (0.95–1.5)	4.6 (4.1–6)	3.4 (3.2–3.9)	<0.0001
SEMA4D, ng/mL	47 (21–81)	32 (31–73)	−14 (−17–11)	0.4459
SEMA7A, ng/mL	0.9 (0.2–1.3)	1.5 (0.78–2.7)	0.5 (0.3–1.1)	<0.0001
**COVID-19 vs. HC**	**Sensitivity (95% CI)**	**Specificity (95% CI)**	**AUC (95% CI)**	***p*-Value**
SEMA3A < 15.50 ng/mL	71.25% (62.34%–78.77%)	72.50% (59.75%–82.40%)	0.7953 (0.7188–0.8718)	<0.0001
SEMA3C > 1.00 ng/mL	63.64% (54.30%–72.05%)	66.67% (53.53%–77.64%)	0.7792 (0.7086–0.8498)	<0.0001
SEMA3F > 2.10 ng/mL	97.47% (92.63%–99.33%)	97.5% (89.54%–99.74%)	0.9981 (0.9951–1.0)	<0.0001
SEMA7A > 1.15 ng/mL	63.75% (54.59%–72.01%)	65% (52.01%–76.09%)	0.7339 (0.6591–0.8087)	<0.0001

**Table 4 biomedicines-11-02786-t004:** Comparing serum semaphorin concentrations, CRP, IL-6 and LDH levels in distinction of moderate from severe/critical COVID-19 and critical from moderate/severe COVID-19. Shown are AUCs with 95% CI.

	Sensitivity (95% CI)	Specificity (95% CI)	AUC (95% CI)	*p*-Value
**Moderate vs. severe/critical**				
SEMA3A > 14 ng/mL	62.5% (51.5%–72.3%)	66.6% (49.8%–80.0%)	0.7586 (0.66–0.85)	0.0003
SEMA3C > 1.80 ng/mL	60.3% (49.1%–70.6%)	58.3% (41.7%–73.2%)	0.6612 (0.56–0.76)	0.0242
SEMA3F < 4.35 ng/mL	72.7% (61.9%–81.3%)	70.8% (54.1%–83.3%)	0.739 (0.54–0.84)	0.0008
SEMA7A < 1.23 ng/mL	66.1% (55.1%–75.5%)	58.3% (41.7%–73.2%)	0.7154 (0.62–0.80)	0.0024
CRP < 95 mg/L	60.7% (49.7%–70.7%)	58.3% (41.7%–73.2%)	0.7428 (0.63–0.85)	0.0087
IL-6 < 53 pg/mL	60.5% (47.2%–72.4%)	75.0% (56.7%–87.2%)	0.7428 (0.63–0.85)	0.0025
LDH < 300 IU/L	80.3% (70.3%–87.6%)	70.8% (54.1%–83.3%)	0.779 (0.68–0.86)	<0.0001
**Critical vs. moderate/severe**				
SEMA3A < 13.0 ng/mL	69.6% (58.8%–78.6%)	70.8% (54.1%–83.3%)	0.7865 (0.69–0.87)	<0.0001
SEMA3F > 4.7 ng/mL	63.6% (52.5%–73.4%)	70.8% (54.7%–83.3%)	0.7591 (0.66–0.85)	0.0003
SEMA7A > 2.0 ng/mL	82.1% (72.3%–89.0%)	70.8% (54.1%–83.3%)	0.8177 (0.73–0.89)	<0.0001
CRP > 100 mg/L	50.0% (39.2%–60.7%)	54.1% (37.8%–69.6%)	0.5432 (0.42–0.65)	0.5426
IL-6 > 60 pg/ml	68.4% (55.1%–79.2%)	55.0% (37.2%–71.5%)	0.6447 (0.51–0.77)	0.0719
LDH > 450 IU/L	76.7% (66.4%–84.7%)	58.3% (41.7%–73.2%)	0.7563 (0.66–0.85)	0.0003

## Data Availability

The datasets generated during and/or analyzed during the current study are available from the corresponding author on reasonable request.

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
