# Peer review of "Association of Immune Semaphorins with COVID-19 Severity and Outcomes"

_biomedicines, 2023, doi:10.3390/biomedicines11102786_

Round 1
Reviewer 1 Report
1. Introduction part is lacking the functional details of target, for that, the authors are requested to provide the detailed functional importance of target, and what if the target is inhibited via diagrammatic or detail representations.
2. Authors have mentioned that, around 80 individuals are included for the analysis, and mentioned that the patients who have concomitant bacterial infection, Active malignant disease, pregnancy, and immunosuppression, were excluded for this study, authors requested to provide the total number of populations have been used for the current study, more in detail.
3. The detailed abbreviations of some short forms are not provided in the manuscript and it makes reading the manuscript difficult.
4. Authors have mentioned that patients with critical and severe COVID-19 had significantly higher serum concentrations of CRP and IL-6, glucose, urea, AST, LDH, and CK at the time of admission. These increased levels of CRP, IL-6 and LDH are related to comorbidities of the above-mentioned category of patients.
5. Does the increased level of SEMA3A, SEMA3C, SEMA3F, SEMA4D and SEMA7A influence the production of and activation of B and T cells? Did authors have checked the impact of T and B cells?
6. In Figure 1b authors have only depicted that sensitivity of Serum concentrations of semaphorin of SEMA3A, SEMA3C, SEMA3F, SEMA4D and SEMA7A, if authors included the other subclasses comparatively, it could be more informative and understandable.
7. SEMA4D is responsible for the activation of immunoregulatory functions that promotes both B cell and DC activation to induce antibody production and antigen-specific T cell responses, in this study, both healthy controls and covid-19 patients show similar concentration of SEMA4D. Did authors predict significant correlation, provide detailed interpretation.
8. Authors have mentioned that non-survivors had significantly lower SEMA3A (11 ng/mL), does the increased level of SEMA3A influence the severity of the COVID-19 condition?
Minor Edits Required
Author Response
- Introduction part is lacking the functional details of target, for that, the authors are requested to provide the detailed functional importance of target, and what if the target is inhibited via diagrammatic or detail representations.
Author's response: We thank the reviewer for this comment. We have mentioned in the introduction the role of semaphorins in various diseases that have been reviewed in other studies to which we have referred readers. A detailed explanation of the semaphorin functional importance is still to be elucidate and we believe is out of the scope of this paper. In the revised version we have more clearly referred readers to references describing their function.
- Authors have mentioned that, around 80 individuals are included for the analysis, and mentioned that the patients who have concomitant bacterial infection, Active malignant disease, pregnancy, and immunosuppression, were excluded for this study, authors requested to provide the total number of populations have been used for the current study, more in detail.
Author's response: This was part of the prospective study (COVID-FAT, ClinicalTrials.gov Identifier: NCT04982328) that had strict inclusion and exclusion criteria to avoid selection bias. We performed a power analysis which is included in the manuscript. The main aim of this part of the study was to evaluate the difference in serum semaphorins concentrations in COVID-19 patients compared to healthy controls and its association with COVID-19 severity. Therefore, 80 consequently hospitalized COVID-19 patients were included. We have added a brief explanation of the study aims. For the same reason we do not believe study flow-chart is needed.
- The detailed abbreviations of some short forms are not provided in the manuscript and it makes reading the manuscript difficult.
Author's response: We have added complete names for indicated abbreviations.
- Authors have mentioned that patients with critical and severe COVID-19 had significantly higher serum concentrations of CRP and IL-6, glucose, urea, AST, LDH, and CK at the time of admission. These increased levels of CRP, IL-6 and LDH are related to comorbidities of the above-mentioned category of patients.
Author's response: We thank the reviewer for this excellent observation. Unfortunately, the purpose of our study was not to investigate how comorbidities affected inflammatory markers and semaphorin serum concentrations. The statistical analysis of such data is limited by the small number of patients and the number of variables. However, the goal of this study was to describe semaphorin serum concentrations in various clinical presentations of COVID-19 and we have revealed it for the first time. In the revised version of the manuscript we have added this in study limitations.
- Does the increased level of SEMA3A, SEMA3C, SEMA3F, SEMA4D and SEMA7A influence the production of and activation of B and T cells? Did authors have checked the impact of T and B cells?
Author's response: That's a good point, unfortunately, we haven't analyzed that. But that's the investigation that should be done in the future to fully comprehend the role of semaphorins in the pathogenesis of COVID-19. This is now added in study limitations.
- In Figure 1b authors have only depicted that sensitivity of Serum concentrations of semaphorin of SEMA3A, SEMA3C, SEMA3F, SEMA4D and SEMA7A, if authors included the other subclasses comparatively, it could be more informative and understandable.
Author's response: We only examined the semaphorins that have been presented.
- SEMA4D is responsible for the activation of immunoregulatory functions that promotes both B cell and DC activation to induce antibody production and antigen-specific T cell responses, in this study, both healthy controls and covid-19 patients show similar concentration of SEMA4D. Did authors predict significant correlation, provide detailed interpretation.
Author's response: We thank the reviewer for this comment. The purpose of this study was to determine whether serum semaphorin concentrations correlate with the severity of COVID-19. SEMA 4D showed no correlation with disease severity. It is unexpected and remains an open question for further research. In the revised version of the manuscript, we comment it in more detail.
- Authors have mentioned that non-survivors had significantly lower SEMA3A (11 ng/mL), does the increased level of SEMA3A influence the severity of the COVID-19 condition?
Author's response: We thank reviewer for this comment. As we mentioned in the discussion serum concentrations of SEMA3A decreased with COVID-19 severity, and patients with higher SEMA3A concentrations had milder forms of the disease. It seems that the role of SEMA3A in the pathogenesis of infections might depend on the receptor utilized which has been the subject of several experimental studies. Inhibition of the SEMA3A/PLXN4 complex is associated with decreased septic response and improved survival rates. In contrast, inhibition of the SEMA3A/NRP1 complex demonstrated increased production of proinflammatory cytokines and higher mortality. In transcriptome study SEMA3A was identified as the most downregulated gene in ARDS patients and overexpression in LPS induced ARDS model alleviates oxidative stress and inflammation. Since the fine-tuned immune response is vital in determining the outcome of COVID-19, we can hypothesize that decreased serum concentration of SEMA3A in COVID-19 patients lead to uncontrolled inflammatory cascade, and in patients with higher concentrations of SEMA3A inflammatory cascade is less pronounced.

Reviewer 2 Report
Your study is a useful contribution to the field, however, you have completely omitted the role of small noncoding RNAs in COVID-19 disease severity. Also you have not mentioned the role of miRNAs in the regulation of semaphorins, see just one example Role of microRNAs in Semaphorin function and neural circuit formation - ScienceDirect
May I suggest a comprehensive revision of the Discussion section which includes the extensive literature on miRNAs in COVID-19 disease severity and their role in the regulation of semaphorins.
Also clearly indicate which COVID-19 variant was active during your study. See this link Age Profiles for SARS-CoV-2 Variants in England and Wales and States of the USA (2020 to 2022): Impact on All-Cause Mor-Tality[v1] | Preprints.org
Hopefully you will be able to transform a good study into an excellent one by wider discussion of the issues.
Author Response
- Your study is a useful contribution to the field, however, you have completely omitted the role of small noncoding RNAs in COVID-19 disease severity. Also you have not mentioned the role of miRNAs in the regulation of semaphorins, see just one example Role of microRNAs in Semaphorin function and neural circuit formation.
May I suggest a comprehensive revision of the Discussion section which includes the extensive literature on miRNAs in COVID-19 disease severity and their role in the regulation of semaphorins.
Author's response:
We thank the reviewer for this comment. Indeed, microRNAs regulating both COVID-19 and semaphorins are important research questions. Unfortunately, here we did not study the impact of microRNAs on semaphorin expression and function during COVID-19. Therefore, we do not think extensive literature review on this topic is needed and might even been confusing for readers. However, we agree that this should be briefly mentioned and have added the following paragraph in the discussion:
“Recent research has shown that micro RNAs (miRNA) control SEMA signaling in immune, cardiovascular, nervous system, and malignancies, directly through their re-ceptors, or indirectly by modulating the molecules that regulate the expression of semaphorins [48]. Similarly, there are reports that specific miRNA signatures in blood or respiratory samples can distinguish COVID-19 disease severity or COVID-19 patients from healthy people [49]. Some of them are linked with semaphorin signaling such as miR-17-5p [50], miR-142-5p [51], miR-126-3p [52], miR-19b-3p [53], miR-92a-3p [54] and miR-320a [55], thus highlighting the potential importance of semaphorin signaling in COVID-19. Interestingly, SARS-CoV2 encoded miRNAs were shown not only to target viral genome and alter viral fitness but can also be transported to host cells during viral infection and bind to host miRNAs and genes and alter immune responses [56]. However, their role in regulating semaphorins remain to be elucidated.”
- Also clearly indicate which COVID-19 variant was active during your study. See this link Age Profiles for SARS-CoV-2 Variants in England and Wales and States of the USA (2020 to 2022): Impact on All-Cause MorTality[v1] | Preprints.org
Author's response: The delta SARS-CoV-2 variant was predominant during our study. This is now added in Study design section.

Round 2
Reviewer 1 Report
The revised version recommends to get accepted
Best wishes to the authors
Reviewer 2 Report
Thank you for your response which I accept.